# Effects of Back Touching on Tidal Volume

**Taichi Hitomi [1,2,*], Chigusa Theresa Yachi [3]** 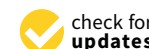 **and Hajime Yamaguchi [4,*]**

1   International Humanities and Social Sciences, J.F. Oberlin University, Tokyo 194-0294, Japan
2   Faculty of Health Science Technology, The Occupational Therapy Department, Bunkyo Gakuin University, Saitama 356-8533, Japan
3   International Humanities and Social Sciences, The International Mental Fitness Institute, J.F. Oberlin University, Tokyo 194-0294, Japan
4   College of Arts and Sciences, J.F. Oberlin University, Tokyo 194-0294, Japan
*   Correspondence: t-hitomi@bgu.ac.jp (T.H.); y-hajime@obirin.ac.jp (H.Y.); Tel.: +81-03-3814-1661 (T.H.)

**Abstract:** The purpose of this basic experiment was to examine the effects of soft touching on an experiment participant's back on tidal volume (TV), as an increase in TV was considered an indication of enhanced relaxation. Healthy experiment participants were divided into an intervention group, where soft touching was administered for two minutes on their back, and a control group, where they were asked to rest. Then the change in TV was measured using a spiro-meter two factor analysis of variance (ANOVA; mixture design) was conducted. As a result of two factor ANOVA, the intervention group's TV changed with statistical significance, while no statistically significant change was observed in the control group. There was a possibility that soft touching on the back had a positive effect on the increase of TV and relaxation. As a result of soft touching on the back, TV was increased. Subjective indicators suggested that the relaxation was enhanced by soft touching on the back.

**Keywords:** touching on the back; tidal volume; relaxation

## 1. Introduction

### 1.1. Effect of Touching on a Back

In the medical fields, there has been some research into the relaxation effects of touching on the back using indices of autonomic nerve system functions [1]. In a preceding study, the relaxation effects of massage on the back as a part of nursing work was examined with patients who had cancer and were receiving chemotherapy. The intervention group exhibited a decrease in anxiety with statistical significance [2]. There was another study done on rats. One group of rats were awake, and the other group of rats were anesthetized and unconscious. Tactile stimuli was given to both sides of the torso of the rats for 5 min, and dopamine, which increases when pleasant and happy feelings are experienced, increased at a statistically significant level on both awake and anesthetized rats [3]. According to the preceding studies, touching on the back was found to have both physically and psychologically positive effects. Therefore, it was presumed valuable to further examine the effects of touching on the back.

### 1.2. Relationship between Breathing and Stress

In the field of physical therapy, hand massage and breathing exercises are administered by a physical therapist to increase tidal volume (TV) [4]. It is believed that an increase in TV enhances relaxation [5]. Some indices were used to examine the physiological and psychological effects of breathing exercises. For example, Lehret et al. [6] claimed that deep breathing was achieved and respiratory sinus arrhythmia (RSA) was enhanced u biofeedback exercise. Lehret et al. [6] posited that

deep breathing possibly enhanced RSA and homeostatic autonomic reflex mechanisms. It was indicated that biofeedback breathing exercises enhanced relaxation and decreased stress level. In addition, Porges [7] claimed that enhanced RSA would possibly indicate the increase of parasympathetic nerve system functions, which would inhibit excessive activation of the sympathetic nerve system. Porges [7] further argued that these functions could possibly be a very important factor in social communication building. Terai et al. [8] examined the stress reduction effect of self-controlled breathing upon both physiological and psychological reactivity against stress stimuli. Terai et al. [8] reported that self-controlled breathing possibly enhanced TV and suppressed increase in heartbeat, which indicated that the cardiac vagus nerve system activities did not decrease. The cardiac vagus nerve activities would normally decrease as a stress reaction. Therefore it was indicated that the stress reaction was mitigated by self-controlled breathing exercises [9]. It is often observed that breathing is used as a somatic approach to control one's psychological state, such as mindfulness breathing [10].

However, it was pointed out that breathing exercises by bio-feedback might have their own risks, as they could cause hyper ventilation [9]. More research shall be conducted to study individual differences in TV, to better benefit the health of people in general.

### 1.3. Tidal Volume

TV, in general, is a volume of air of one breath (the sum of inhalation and exhalation). The average TV of healthy adults is approximately 500 mL per breath. The average number of breaths per minute for healthy adults is approximately 12. Therefore the average TV is said to be approximately 6 L/min. TV may vary from time to time, and may not be 500 mL all the time. The breathing pattern has variations depending on individuals and conditions [11,12]. For example, if an individual takes a deep breath, TV increases and number of respiration decreases. If an individual engages in a physical exercise, both TV and rate of respiration increases. In addition, the health effects of increased tidal volume have also been reported. According to Vlemincx et al. [13], lungs stretch by sighing and ventilation efficiency rises. Also, according to Inoue et al. [14], a sigh is a breath of at least twice the regular TV. A preceding study reported that such deep breaths help to reduce muscle tension and subjective stress-related feelings, and are closely related to balance in the autonomic nerve system and maintenance of homeostasis [15]. Another study reported that the TV of natural breathing is related to the heart beat, and the relationship was examined using heart rate variability bio-feedback method. The study claimed that slow breathing was related to an enhancement of respiratory sinus arrythmia and helped in regulating the autonomic nervous system [9]. Therefore, it can be said that an increase in TV is expected to have a positive impact on health.

### 1.4. Purpose of the Research

Touching on the back can possibly increase TV and thus enhance relaxation. However, in the preceding studies, there has been no research conducted taking TV as an index. Therefore, in this study, the effect of touching on a back was examined using TV as an index.

Considering comfortable tactile stimulus, C-tactile fibers play an important role. C-tactile fibers are reported to enhance relaxation of the mind and the body, and the optimum stroking can be beneficial [16]. C-tactile fibers are most activated by smooth and soft touching [17,18]. C-tactile fibers are less activated by a quick motion of touching. Quick and rough touching increased the symptoms of restless legs syndrome [19]. It is necessary to consider the best touching method to effectively stimulate C-tactile fibers.

## 2. Materials and Methods

### 2.1. Experiment Participants and Ethical Consideration

This study was basic research, therefore healthy adults of both male and female were selected as experiment participants. A document was shown to the experiment participants which explained

that it would be voluntary to participate in the experiment and there would not be any negative consequence if they decide not to participate. The document was also verbally read to the experiment participants. Once they expressed their agreement, they were chosen as experiment participants. The research was approved by the research ethics committee of Tokorozawa Rehabilitation hospital where the experiment participants were working (date of approval: 30 September 2016).

The purpose of the research was to examine the change in TV after touching on the back of an experiment participant. Touching was administered while an experiment participant sat on a chair. In this setting, it was impossible to eliminate the effect of resting. To eliminate the effect of resting in a seated position, the participants were divided into an intervention group and a control group.

Those who agreed to participate in the experiments were randomly divided into an intervention group (9 males and 8 females; total 17), and a control group (6 males and 6 females; total 12). The therapist was a male physical therapist with 13 years of working experience. The therapist had not received any training in touching or any license/qualification in the field of touching.

## 2.2. Physiological Index

The promotion of conscious breathing control through the cerebral cortex is considered to be at risk of causing hyperpnea and hyperventilation [9]. According to Sakakibara [9], slow and restful breathing stabilizes autonomic nervous system function. Therefore, this study used TV to focus on natural and unintended respiration changes.

TV was measured using a spirometer ( Autospiro AS-307, Minato Ikagaku company) (unit: L). The spirometer gives instructions to the experiment participant, first, to take a normal breath, to inhale to the maximum extent, exhale to the maximum extent, and lastly, take another normal breath. The spirometer measures the vital capacity, TV, the inspiratory reserve, the expiratory reserve, the inspiratory capacity, and other variables. The spirometer measures air flow using a hot wire flowmeter. The volume of air is determined according to the volume value from the flow meter. The spirometer measures several breaths. It detects the difference of the amount of air per breath (sum of inhalation and exhalation). If the difference of the amount of each breath is smaller than 50 mL, the spirometer starts recording. It records 5 breaths. If the difference of the amount of air of each breath is all less than 50 mL, then, the spirometer takes an average of those relatively even breaths. Then, the spirometer guides the experiment participant to take a deep breath and takes the measurement of forced vital capacity. Once forced vital capacity is measured, the measurement will finish. The Spirometer waits until the difference in amount of air in each natural breath is less than 50 mL. If the amount fluctuates, the spirometer waits for a while until the breaths are relatively even, with less than 50 mL difference. Therefore, the time required for measurement can vary depending on each individual. The spirometer measures forced vital capacity. In this research, we took the TV (average of 5 breaths) as an outcome measure.

## 2.3. Method of Touching and a Therapist

It was reported in a preceding study that parasympathetic nerve system was stimulated when an experiment participant was touched at a speed of 3 to 5 cm per second, which was considered a fairly soft touch [20,21]. It was thought that the result would be that unmyelinated mechanoreceptors reacted, not the myelinated afferents [17]. On the other hand, with touching at a fast speed, it was thought that unmyelinated mechanoreceptors would be less activated [18]. Therefore, a therapist applied gentle touching with both hands from the top to the bottom of the back of an experiment participant at the speed of 3 to 5 cm per second. Touching was performed for two minutes. During touching, an experiment participant was asked to sit on a chair and instructed to relax.

## 2.4. Procedure

An experiment participant was instructed to relax at the time of measurement. The experiment participant was seated throughout the experiment, except at the time of TV measurement by a

spirometer. An experiment participant in the intervention group was seated while receiving touching and an experiment participant of the control group was seated and rested in silence, both for two minutes. To measure TV, an experiment participant was instructed to stand up, wear a nose clip and bite a mouth piece. A voice message was given and the experiment participant breathed in a resting state, and made a deep breath. In addition, the clothes of the subjects were T-shirts.

The experiment was conducted in early October in 2016 and the time of experiment was set as after 18:00. The experiment protocol is described in Figure 1.

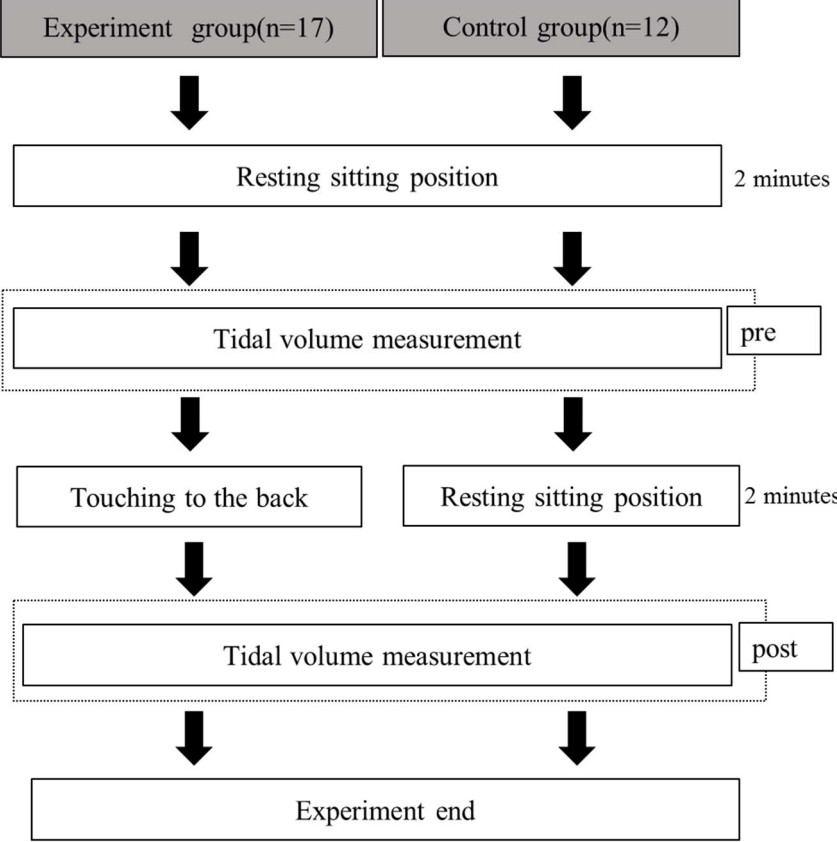

**Figure 1.** Experiment protocol.

*2.5. Statistical Analysis*

Pre- and post-experimental data for both the intervention and the control group were analyzed using two factor analysis of variance (ANOVA; mixture design), and the main effect and the cross effect were determined. If a cross effect was detected, then the simple main effect was examined. The level of statistical significance was set at 5%. HAD Ver16.0 software was used for statistical analysis [22].

## 3. Results

The following are the summary value (Table 1) and the individual data (Table 2).

**Table 1.** Summary Values.

| Variable Name | n | Mean | Median | SD | Varinace | Min | Max |
|---|---|---|---|---|---|---|---|
| Control group-pre | 12 | 0.765 | 0.755 | 0.162 | 0.026 | 0.460 | 1.000 |
| Control group-post | 12 | 0.763 | 0.730 | 0.153 | 0.023 | 0.510 | 1.000 |
| Experiment group-pre | 17 | 0.674 | 0.650 | 0.123 | 0.015 | 0.500 | 1.000 |
| Experiment group-post | 17 | 0.848 | 0.900 | 0.167 | 0.028 | 0.470 | 1.070 |

**Table 2.** Individual data.

| | Control Group | | | | Experiment Group | | |
|---|---|---|---|---|---|---|---|
| | Pre | Post | Post-pre | | Pre | Post | Post-pre |
| 1 | 0.96 | 0.72 | −0.24 | 1 | 0.69 | 0.93 | 0.24 |
| 2 | 0.85 | 1.00 | 0.15 | 2 | 0.75 | 0.99 | 0.24 |
| 3 | 0.78 | 0.95 | 0.17 | 3 | 0.62 | 1.01 | 0.39 |
| 4 | 0.70 | 0.71 | 0.01 | 4 | 0.60 | 0.90 | 0.30 |
| 5 | 1.00 | 0.64 | −0.36 | 5 | 0.68 | 0.67 | −0.01 |
| 6 | 0.71 | 0.61 | −0.10 | 6 | 0.52 | 0.84 | 0.32 |
| 7 | 0.46 | 0.91 | 0.45 | 7 | 0.61 | 0.85 | 0.24 |
| 8 | 0.65 | 0.90 | 0.25 | 8 | 1.00 | 0.94 | −0.06 |
| 9 | 0.59 | 0.64 | 0.05 | 9 | 0.77 | 1.07 | 0.30 |
| 10 | 0.75 | 0.51 | −0.24 | 10 | 0.65 | 0.60 | −0.05 |
| 11 | 0.97 | 0.82 | −0.15 | 11 | 0.65 | 0.73 | 0.08 |
| 12 | 0.76 | 0.74 | −0.02 | 12 | 0.50 | 0.76 | 0.26 |
| | Mean | | 0.00 | 13 | 0.75 | 0.99 | 0.24 |
| | | | | 14 | 0.80 | 0.71 | −0.09 |
| | | | | 15 | 0.65 | 0.95 | 0.30 |
| | | | | 16 | 0.50 | 0.47 | −0.03 |
| | | | | 17 | 0.71 | 1.01 | 0.30 |
| | | | | | Mean | | 0.17 |

By two factor ANOVA, the pre-post main effect (F (1, 25) = 5.57, p = 0.03), and the cross effect (F (1, 25) = 5.89, p = 0.02) was obtained (Table 3).

**Table 3.** Results of two-way analysis of variance.

| Variable Name | Partial η2 | 95% CI | F Value | P Value |
|---|---|---|---|---|
| Touching terms | 0.00 | 0.000, 0.048 | 0.00 | 0.948 |
| Pre-Post | 0.17 | - | 5.57 | 0.026 * |
| Interaction | 0.18 | - | 5.89 | 0.022 * |

* p < 0.05.

As the cross effect exhibited statistical significance, the simple main effect was examined. As a result, the statistical significance was observed in the intervention group (F (1, 25) = 13.85, p = 0.00), while no statistical significance was observed in the control group. Therefore, it was indicated that touching on the back possibly increased TV in the experiment participants (Figure 2, Table 4).

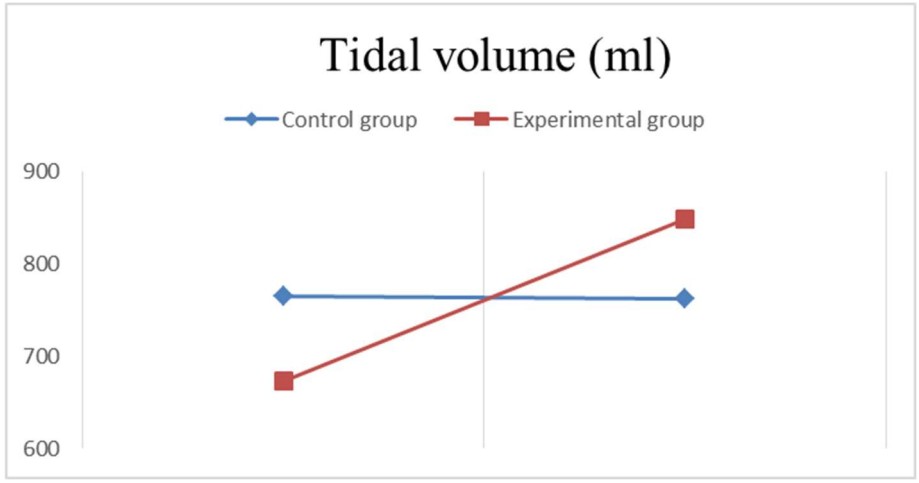

**Figure 2.** Comparison of experimental group and control group.

**Table 4.** Simple effect of factors.

| Variable Name | Standard Error | Effect Size d | 95% CI | t Value | P Vaule |
|---|---|---|---|---|---|
| Experiment group (pre)-control group (pre) | 0.057 | 1.161 | 0.129, 2.194 | 1.604 | 0.114 |
| Experiment group (post)-control group (post) | 0.057 | −0.794 | −1.542, −0.045 | 1.504 | 0.138 |
| Experiment group (pre)-Experiment group (post) | 0.047 | −1.115 | −1.820, −0.411 | 3.721 | 0.001 ** |
| control group (pre)-control group (post) | 0.056 | 0.022 | −1.045, 1.089 | 0.045 | 0.965 |

** $p < 0.01$.

## 4. Discussion

This study examined the relaxation effect of touching on the back, reflected in TV. An experiment participant was seated while touching was administered. It was impossible to eliminate the factor of resting in a seated position. There had to be a way to obtain the effect of touching, free from the effect of resting in a seated position. Therefore, the experiment participants were divided into an intervention group with touching, and a control group without touching. The comparison was made using two factor ANOVA.

As a result, statistical significance was obtained in both the main and cross effects in the intervention group. Thus, multiple comparison by Holm method was conducted. Concerning the increase in TV, there was no statistical significance obtained in the control group, whereas statistical significance was obtained in the intervention group. This would indicate the possibility that the TV was increased by the touching employed in this research. It was first expected that TV might increase just by resting in a seated position in the control group. However, the results of the experiment showed that TV did not increase just by resting. Therefore, it is possible that the touching on the back encouraged the experiment participants to breath deeper. Thus, the increase of TV in this experiment would be considered appropriate, by looking at the statistical summary of the difference between pre and post touching. Hayano et al. [23] claimed that RSA as an index of function of sympathetic nerve system would be significantly increased by the increase of TV. If rate of breathing decreases and TV increases, alveolar volume and energy efficiency will be increased. The touching in this experiment possibly increased RSA, which was an indicator of enhanced function of the parasympathetic nerve system.

The change of TV indicates that both physical and psychological relaxation was possibly achieved by touching. This was achieved by slow touching, which theoretically activated the C-tactile fibers [24]. Furthermore, there were a few preceding studies indicating that there would be some relationship between TV and stress [25]. In this study, an open ended question was asked to the experiment participants of the experiment group. The following comments were made: "I can easily breath"; "My body feels light"; "I feel comfortable"; "My body feels heavy"; and "I feel more sleepy". The physiological data and the subjective data matched, as both of them indicated that the experiment participants experienced relaxation and a calming state. Matsutani el al [15] reported that deep breath helped to release muscle tension. Deep breath was related to balance in the autonomic nerve system and was considered to help the body to maintain homeostasis. Vlemincx et al. also reported that once an experiment participant experienced release of stress, she would feel relief as a subjective observation, and there was a voluntary deep breath [13,26]. If TV was increased, it meant that an experiment participant was taking a deeper breath. It is possible that the experiment participant experienced stress reduction and relaxation while taking a deeper breath. It will be meaningful to examine the effects of touching on stress.

## 5. Limitations and Future Development

The purpose of this study was to examine the physiological and psychological effects of touching, and this research was not designed to examine any respiratory diseases. In this research, it became

clear that touching possibly enhanced TV. However, other factors were not considered in combination with TV. In the future, the autonomous nerve system index, and other psychological indexes, such as questionnaires, shall be employed to determine the relaxation effect of touching. TV is but one of many factors to determine the relaxation state of experiment participants. Furthermore, touching methods shall be examined. Strength of pressure, speed and direction of touching shall be examined to obtain the optimum touching method in the future. In this research, an open-ended question was not asked to the control group. Subjective data is an important information for analysis. Therefore, in the future, such subjective data should be collected and analyzed.

## 6. Conclusions

This study examined the relaxation effect on TV of touching on the back for two minutes. It became clear that TV was increased by touching on the back. TV showed possible enhancement of relaxation through soft touching on the back. It is presumed that touching on the back possibly has the power to mildly control the breathing of an experiment participant, and therefore to help him/her achieve relaxation.

**Author Contributions:** Conceptualization, T.H. and H.Y.; methodology, T.H.; software, T.H.; validation, T.H.; formal analysis, T.H.; investigation, T.H.; resources, T.H.; data curation, T.H.; writing—original draft preparation, T.H. and C.T.Y.; writing—review and editing, T.H. and C.T.Y.; visualization, H.Y.; supervision, H.Y.; project administration, T.H.

**Funding:** This research received no external funding.

**Conflicts of Interest:** The authors declare no conflict of interest.

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
