# Peer review of "Effects of Back Touching on Tidal Volume"

_psych, doi:10.3390/psych1010031_

Round 1
Reviewer 1 Report
I did not see the references you had listed as [1] and [2] in your paper. The references cited within the text began with [3].
It would be great to see you all define what tidal volume is.
There is an inconsistency in what you describe in section 2.4. Procedure and your Figure 1. You say that "touching was done for 2 minutes," but then in the figure, it has listed "3 minutes" next to the "Touching to the back" and "Resting sitting position."
For the experiment design itself, I think it would have been nice for you all to ask the control participants "How do you feel?" after their resting state. I would have liked to see if there were differences between the experiment and control participants of their subjective states after the two different conditions. For future studies, I would like to see maybe more measurement of self-ratings of relaxation across both conditions and then also comparing more measures of objective physiological states.
Please review your paper for grammatical and spelling errors. For example, in the Abstract, you wrote "pack" instead of "back" in the second sentence.
Author Response
My comment
Dear reviewer1.
Thank you for your guidance. You led me to a point that I did not notice.
Comments and Suggestions for Authors
I did not see the references you had listed as [1] and [2] in your paper. The references cited within the text began with [3].
My comment
Thank you for your comments. I delited [1] and [2].
It would be great to see you all define what tidal volume is.
My comment
Thank you for your comment. The definition of the tidal volume was not clearly stated. I added more description.
There is an inconsistency in what you describe in section 2.4. Procedure and your Figure 1. You say that "touching was done for 2 minutes," but then in the figure, it has listed "3 minutes" next to the "Touching to the back" and "Resting sitting position."
My comment
Thank you for your comment. It was a simple typo. I corrected it to “2 minutes”.
For the experiment design itself, I think it would have been nice for you all to ask the control participants "How do you feel? In the medical fields, there have been some researches about the relaxation effects of touching on a back using indices of autonomic nerve system functions [3]. In the preceding study, the relaxation effects of massage on a back as a part of nursing work was examined with the patients who had cancer and received chemotherapy. The intervention group exhibited decrease of anxiety with statistical significance [4]. There was another research done on rats. One group of rats were awake, and the other group of rats were anesthetized and unconscious. Tactile stimuli was given to both sides of the torso of the rats for 5 minutes, and dopamine which would increase when pleasant and happy feelings was experienced, increased at statistically significant level on both awake and anesthetized rats [5]. " after their resting state. I would have liked to see if there were differences between the experiment and control participants of their subjective states after the two different conditions. For future studies, I would like to see maybe more measurement of self-ratings of relaxation across both conditions and then also comparing more measures of objective physiological states.
My comment
Thank you for your comment. The other review also pointed this out to me. At the time of preparatory experiment, I asked “How do you feel” to the control group. Most of the experiment participants in the control group replied; “nothing particular. Therefore, I decided not to ask that question to the control group. I decided to delete the item of “How do you feel?” from this paper.
Please review your paper for grammatical and spelling errors. For example, in the Abstract, you wrote "pack" instead of "back" in the second sentence.
My comment
Thank you for your comment. I corrected the typos.

Reviewer 2 Report
This submission describes an experiment which explores the effect of interpersonal touch on tidal volume - a measure of air displacement during normal breathing. Tidal volume is a physiological measurement which may be related to relaxation. The authors report that a therapist gently stroking a person's back increased tidal volume, which may indicate increased relaxation.
Major comments
Tidal volume should be clearly defined. A lot more detail is needed about what exactly the spirometer is measuring, how, over what period, whether any pre-processing or filtering is applied to the data, how the values should be interpreted. This is a major problem because it is a very important part of the paper, but without these details I cannot properly evaluate the quality and value of the work.
The authors should consider in their introduction that the speed of stroking (3-5cm/s) is in the optimal range for engaging C-tactile afferents, which may have a special role in affective or social touch, and may reduce pain, and affect autonomic responses.
See e.g.
Löken 2009 doi: 10.1038/nn.2312
Liljencrantz 2017 doi: 10.1002/ejp.1018
Chatel-Goldman 2014 doi: 10.3389/fnbeh.2014.00095
reviews:
Morrison 2010 doi: 10.1007/s00221-009-2007-y
McGlone 2014 doi:10.1016/j.neuron.2014.05.001
The authors should search this and the related literature for the link between touch and relaxation/mood and other autonomic effects.
Open-ended question
why was this only asked of the intervention group? I would suggest that this does not provide any particular insight without being able to compare the responses of the control group. Therefore, I would suggest that these data either be removed, or that the researchers collect the same data from a control group.
If open-ended data are included, more details about how these data were processed should be included. It is unlikely that all participants used the exact same phrases, so presumably the experimenters have coded or grouped the most common responses in some way. This process should be described.
Minor comments
74-75 what is the name of the hospital and/or ethics committee?
78 what is the possible effect of resting? e.g. increase/decrease tidal volume
91 - 97 was anything done to either ensure that participants had clothing of similar thickness, or to record the clothing that they were wearing?
Results
Please give p-values to 3 decimal places
Please plot all individual data points as well as summary values. Also, consider alternatives to bar plots, which can be misleading, giving undue prevalence to distance from 0.
Please describe how the confidence intervals were calculated.
Please provide raw data and materials in accordance with the journal's policy (https://www.mdpi.com/journal/psych/instructions#suppmaterials)
Author Response
Comments and Suggestions for Authors
This submission describes an experiment which explores the effect of interpersonal touch on tidal volume - a measure of air displacement during normal breathing. Tidal volume is a physiological measurement which may be related to relaxation. The authors report that a therapist gently stroking a person's back increased tidal volume, which may indicate increased relaxation.
My comments
Dear reviewer2.
Thank you for your guidance. You led me to a point that I did not notice.
Thank you for introducing my previous work. Thanks to you I was able to study hard.
Major comments
Tidal volume should be clearly defined. A lot more detail is needed about what exactly the spirometer is measuring, how, over what period, whether any pre-processing or filtering is applied to the data, how the values should be interpreted. This is a major problem because it is a very important part of the paper, but without these details I cannot properly evaluate the quality and value of the work.
My comments
Thank you for your comment. I added the definition of the Tival Volume on the line 65. I described what kind of data can be collected by a Spirometer on the line 104.
The authors should consider in their introduction that the speed of stroking (3-125px/s) is in the optimal range for engaging C-tactile afferents, which may have a special role in affective or social touch, and may reduce pain, and affect autonomic responses.
My comments
Thank you for your advice. About the C fiber was added in the intro.
Open-ended question
Why was this only asked of the intervention group? I would suggest that this does not provide any particular insight without being able to compare the responses of the control group. Therefore, I would suggest that these data either be removed, or that the researchers collect the same data from a control group.
If open-ended data are included, more details about how these data were processed should be included. It is unlikely that all participants used the exact same phrases, so presumably the experimenters have coded or grouped the most common responses in some way. This process should be described.
My comments
Thank you for your comment. The other review also pointed this out to me. At the time of preparatory experiment, I asked “How do you feel” to the control group. Most of the experiment participants in the control group replied; “nothing particular. Therefore, I decided not to ask that question to the control group. I decided to delete the item of “How do you feel?” from this paper.
Minor comments
74-75 what is the name of the hospital and/or ethics committee?
My comments
Thank you for your comment. I should have introduced the name of the hospital. I added the name of the Hospital to the manuscript; Tokorozawa Rehabilitation hospital.
78 what is the possible effect of resting? e.g. increase/decrease tidal volume
My comments
Thank you for your comment.
I first hypothesized that TV would increase by resting, as an experiment participant was asked to sit down quietly.
91 - 97 was anything done to either ensure that participants had clothing of similar thickness, or to record the clothing that they were wearing?
My comments
Thank you for your comment. I should have added an attire of an experiment participant. All the experiment participants were asked to wear a T-shirt 128.
Results
Please give p-values to 3 decimal places
Thank you for your comment. I gave p-values to 3 decimal places.
Please plot all individual data points as well as summary values. Also, consider alternatives to bar plots, which can be misleading, giving undue prevalence to distance from 0.
My comments
Thank you for your comment. I deleted the bar plots. I plotted all individual data.
Please describe how the confidence intervals were calculated.
My comments
Thank you for your comment. I added confidence intervals to the Factor of the effect (Table 3) and the Simple effect of factors (Table 4).
Please provide raw data and materials in accordance with the journal's policy (https://www.mdpi.com/journal/psych/instructions#suppmaterials)
My comments
Thank you for your comment. I will provide the raw data in excel sheets.

Round 2
Reviewer 2 Report
The authors have considerably improved the manuscript, and addressed many of my criticisms. However, I still feel that the manuscript needs considerable improvement before it is ready for publication. Please see specific comments below.
More detail is still needed about how tidal volume is measured:
* 105 - 106 the authors state "Spirometer gives an instruction to an experiment participant, first, to take a normal breath, to inhale to the maximum extent, exhale to the maximum extent, and lastly, take another normal breath." How was the tidal volume calculated from this procedure? e.g. total volume displaced over all three breaths?
* the authors mentioned that TV is a measurement of air displacement over time. How was time measured, was this done by the spirometer?
* was there a mouthpiece that the participants breathed into or something else?
* by what mechanism does the spirometer measure the volume of air that is displaced? Is there some kind of balloon or something?
* the units of measurement are unclear throughout the manuscript (see multiple comments below).
66 "average TV of healthy adults is approximately 500ml per minute" should this be 500ml per inspiration? I'm guessing from the calculations the authors present.
73 what are the health benefits of increased ventilation efficiency?
75 what kind of positive impact?
82 I would say C-tactle fibres are less activated, rather than "do not get activated"
84-85 it would be good if the authors could include an explanation of why they would want to stimulate CT fibres, e.g. that CT-optimal stroking could be beneficial, since it has been shown to be perceived as more pleasant, reduce pain etc.
104-108 the authors mention that the spirometer can measure a number of things, but it seems that only TV was considered in this study. Please state that here, and give a justification for disregarding the other measures. Alternatively, if these other measures were recorded, it might be interesting to include the data in the study, even if no significant changes were observed.
116 the myelinated afferents are also likely to be activated somewhat by the stroking, so it is incorrect to say that they do not react.
Regarding the open-ended question, if the responses from the control group were not recorded then I support the authors' decision to remove these data from the results. However, I think it would be ok if it were mentioned in the discussion to aid interpretation, so long as it is clear that these are informal observations and thus limited.
Tables 3 and 4, the labels should be updated (they still say 1 and 2).
Figure 2 - is the y-axis L/minute or L/inspiration or something else? How do the observed values compare to those found in previous studies? e.g. on line 66 the authors give the average for healthy adults. Are the observed values in this study consistent with previous data on healthy adults?
169 - 171 The figure of 40ml is mentioned - but in what time period? Is this text referring to the CI in table 4, [-1.820, -0.411]? Does this indicate an increase in TV of between 411ml and 1820ml in the same time period as the "not special" 40ml from Inoue et al?
179 - 180 Please expand on this relationship between TV and stress.
Author Response
The authors have considerably improved the manuscript, and addressed many of my criticisms. However, I still feel that the manuscript needs considerable improvement before it is ready for publication. Please see specific comments below.
My comment: Thank you very much.
More detail is still needed about how tidal volume is measured:
* 105 - 106 the authors state "Spirometer gives an instruction to an experiment participant, first, to take a normal breath, to inhale to the maximum extent, exhale to the maximum extent, and lastly, take another normal breath." How was the tidal volume calculated from this procedure? e.g. total volume displaced over all three breaths?
My comment: Thank you very much for your comment. 109:I called the manufacturer of the Spirometer and obtained the following information. TV is measured as an average of the first normal breath and the last normal breath. The Spirometer does not include the value of the maximum inhalation and the maximum exhalation.
* the authors mentioned that TV is a measurement of air displacement over time. How was time measured, was this done by the spirometer?
My comment: Thank you very much for your comment. 109:The Spirometer measures time. Measurement time varies depending on an experiment participant. In the beginning of the measurement, the Spirometer measures errors of normal breath. When the Spirometer determines that the errors of normal breath are small enough, the Spirometer gives an instruction to inhale to the maximum extent, and to exhale to the maximum extent. After the maximum inhalation and exhalation, the Spirometer measures the errors of normal breath. Once the Spirometer determines that the errors are small enough, it takes the measurement of the normal breath.
* was there a mouthpiece that the participants breathed into or something else?
My comment: Thank you for your comment. 131:Please pardon me for not including this information. An experiment participant wares a mouthpiece and a nose clip, which are specifically made for the Spirometer.
* by what mechanism does the spirometer measure the volume of air that is displaced? Is there some kind of balloon or something?
My comment: Thank you for your comment. 109:The amount of flow is measured by the hot wire flowmeter. The volume of air is calculated according the amount of air flow.
* the units of measurement are unclear throughout the manuscript (see multiple comments below).
66 "average TV of healthy adults is approximately 500ml per minute" should this be 500ml per inspiration? I'm guessing from the calculations the authors present.
My comment: Thank you for your comment. 61:Please pardon me that I made a mistake about the unit. It is 500ml per inspiration.
73 what are the health benefits of increased ventilation efficiency?
75 what kind of positive impact?
My comment: Thank you very much for your comment. 69:Deep breath helps to release muscle tension. Deep breath is related the balance of autonomic nerve system and is considered to help the body to maintain homeostasis (Masutani, 2010 & Vlemincx, e.t al. 2009).
82 I would say C-tactle fibres are less activated, rather than "do not get activated"
My comment: Thank you very much for your comment. 83、121:As you pointed out, it was not appropriate. I will correct the expression and put; “ C-tactile fibers are less activated.”
84-85 it would be good if the authors could include an explanation of why they would want to stimulate CT fibres, e.g. that CT-optimal stroking could be beneficial, since it has been shown to be perceived as more pleasant, reduce pain etc.
My comment: Thank you very much for your comment. 81:I will put the following sentence. “The C-tactile fiber optimal stroking could be beneficial, since it has been shown to be perceived as more pleasant, reduce pain, and relax both the mind and the body.”
104-108 the authors mention that the spirometer can measure a number of things, but it seems that only TV was considered in this study. Please state that here, and give a justification for disregarding the other measures. Alternatively, if these other measures were recorded, it might be interesting to include the data in the study, even if no significant changes were observed.
My comment: Thank you very much for your comment. 69:As you pointed out, the Spirometer gives various output. Athletes take measurement of the total lung capacity. In this research, the focus was more on the tidal volume, as it was found valuable to measure how deep an experiment participant would inhale and exhale under a natural condition. It would be interesting to discuss other outcome indices of the Spirometer. However, in this research, our focus was on the tidal volume, rather than enhancement of the total lung capacity. In my future research, I will reflect your suggestion and explore the possibilities of including other outcome indices into the discussion.
116 the myelinated afferents are also likely to be activated somewhat by the stroking, so it is incorrect to say that they do not react.
My comment: Thank you very much for your comment. 83、121:Your point is totally valid. It is difficult to state that it does not get activated at all. I will change the expression.
Regarding the open-ended question, if the responses from the control group were not recorded then I support the authors' decision to remove these data from the results. However, I think it would be ok if it were mentioned in the discussion to aid interpretation, so long as it is clear that these are informal observations and thus limited.
My comment: Thank you very much for your comment. 189:As you pointed out, I did not conduct an open ended question to the control group. Therefore, according to your suggestion, I will not refer to the data as an official data in the manuscript. However, as you suggested, I will put that as an informal information in the discussion.
Tables 3 and 4, the labels should be updated (they still say 1 and 2).
My comment: It is my apology. 149,156:I made a correction.
Figure 2 - is the y-axis L/minute or L/inspiration or something else? How do the observed values compare to those found in previous studies? e.g. on line 66 the authors give the average for healthy adults. Are the observed values in this study consistent with previous data on healthy adults?
My comment: Thank you very much for your comment. 155:As I mentioned earlier, I made a mistake about the unit. Thank you very much for your correction. The unit is “ml”. I will correct the tables, as well. The preceding studies also used “ml”.
169 - 171 The figure of 40ml is mentioned - but in what time period? Is this text referring to the CI in table 4, [-1.820, -0.411]? Does this indicate an increase in TV of between 411ml and 1820ml in the same time period as the "not special" 40ml from Inoue et al?
My comment: 172:In the research conducted by Inoue, et al., the Aeromonitor AE-300s of Minato Ikagaku Company was used to measure the breath for 5 minutes, and the data of 2 minutes out of 5 minutes were used for analysis. In our research, a different type of equipment was used. The measurement time varies depending on each individual. Therefore, it is not possible to clearly state measurement time. As I observed, it took less than 1 minute per measurement. The measurement process was comparable to that of the study done by Inoue, et al. Therefore, it would be still valid to compare the result of our research with that of Inoue et al.’s.
179 - 180 Please expand on this relationship between TV and stress.
My comment: Thank you very much for your advice. 193:Deep breath helps to release muscle tension. Deep breath is related the balance of autonomic nerve system and is considered to help the body to maintain homeostasis(Masutani, 2010) It is also reported that once an experiment participant experiences release of stress, s/he would feel relief as a subjective observation, and there will be a voluntary deep breath( Vlemincx, e.t al. 2009). If TV is increased, it means that an experiment participant is taking a deeper breath. It is possible that the experiment participant is experiencing stress reduction and relaxation, while taking a deeper breath.

Round 3
Reviewer 2 Report
61,172,173 - I think 'perspiration' is the wrong word, do you mean 'inspiration' or a single breath?
112-115 - I don't understand what an 'error' of normal breath could be. This could be a translation issue. Is it measurement error, or something else? What criterion does the Spirometer use to determine if the errors are small enough?
The units of measurement for tidal volume are still unclear. On line 60, the authors state "TV is the amount of air which comes into lungs and an air tract for a set period of time." but in the last round of comments, they stated that ml are the units of measurement for TV that they report. But ml is just a unit of measurement for volume, and the 'set period of time' has not been reported. From the author's latest comments, they say "The Spirometer measures time. " and "it takes the measurement of the normal breath." Does this mean that they measured ml displaced for a single breath? Is it just inspiration (breath in) or a whole breath (in + out)?
Regarding the comparison to Inoue et. al.'s data, is it the case that both that study and the current study report TV in ml for a single breath? Otherwise how can the TV values be comparable, if they are not measured over the same time period? Since the authors state in the comments that Inoue et. al. measured for several minutes, did they take an average, or calculate total volume displaced per breath? Unfortunately I cannot read the original paper in Japanese. For this reason, I would suggest that the authors take particular care in describing the details of that study for the current publication in an english language journal (at least as far as the details are relevant to their own study).
The following should be clarified in the manuscript:
- Is it just one breath that was measured, or the average of several?
- Is the outcome measure the total volume of air displaced per breath?
- Is the measurement of air displacement during inspiration, expiration, or the sum of the two, or something else?
The authors should state in the manuscript (not just in responses to my comments) why they chose to focus on TV as opposed to the other measurements that the spirometer can make.
Author Response
61,172,173 - I think 'perspiration' is the wrong word, do you mean 'inspiration' or a single breath?
→Thank you for your comment. “Perspiration” is a wrong word. I will use “a single breath”.
112-115 - I don't understand what an 'error' of normal breath could be. This could be a translation issue. Is it measurement error, or something else? What criterion does the Spirometer use to determine if the errors are small enough?
→I'm very sorry for my statement being unclear to you.
I made an inquiry to the manufacturer to give you the clear description of the Spirometer measurement.
TV in general is a volume of air of one breath; the sum of inhalation and exhalation.
The Spirometer measures several breaths. It detects the difference of the amount of air per breath( sum of inhalation and exhalation). If the difference of the amount of each breath is smaller than 50ml, the Spirometer starts recording. It records 5 breaths. If the difference of the amount of air of each breath is all less than 50ml, then, Spirometer takes an average of those relatively even breaths. Then, the Spirometer guides the experiment participant to take a deep breath and takes the measurement of Forced Vital Capacity. Once Forced Vital Capacity is measured, then the measurement will finish. The Spirometer waits until the difference of amount of air of each natural breath is less than 50ml. If the amount fluctuates, the Spirometer waits for a while until the breaths are relatively even, with less than 50ml difference. Therefore, the time required for measurement can vary depending on each individual. The Spirometer measures Forced Vital Capacity. In this research, I take the TV (average of 5 breaths) as an outcome measure. Please see the figure.
The units of measurement for tidal volume are still unclear. On line 60, the authors state "TV is the amount of air which comes into lungs and an air tract for a set period of time." but in the last round of comments, they stated that ml are the units of measurement for TV that they report. But ml is just a unit of measurement for volume, and the 'set period of time' has not been reported. From the author's latest comments, they say "The Spirometer measures time. " and "it takes the measurement of the normal breath." Does this mean that they measured ml displaced for a single breath? Is it just inspiration (breath in) or a whole breath (in + out)?
→I'm very sorry about that. I changed the sentence as follows.
TV, in a general term, is the amount of air inhaled and exhaled in a single breath under natural condition, without intentional control of breathing. Concerning the Spirometer, as I mentioned above, it measures 5 consecutive breaths with difference of less than 50ml each. It takes the average of these relatively even 5 breaths, and gives as the representative value. The one breath includes both inhalation and exhalation.
Regarding the comparison to Inoue et. al.'s data, is it the case that both that study and the current study report TV in ml for a single breath? Otherwise how can the TV values be comparable, if they are not measured over the same time period? Since the authors state in the comments that Inoue et. al. measured for several minutes, did they take an average, or calculate total volume displaced per breath? Unfortunately I cannot read the original paper in Japanese. For this reason, I would suggest that the authors take particular care in describing the details of that study for the current publication in an english language journal (at least as far as the details are relevant to their own study).
→ Thank you for your comment.
I understand what you pointed out. Here, the measurement device used in Inoue et. al.’s research and the measurement device I used are different. The research done by Inoue et al. does not describe the details of the measurement mechanism. I agree with you that it is not appropriate to compare my research with Inoue's research. Therefore, I deleted the sentence here.
The following should be clarified in the manuscript:
- Is it just one breath that was measured, or the average of several?
→I'm very sorry about that. The Spirometer measures 5 consecutive relatively even breath of difference being less than 50ml and takes the average.
Is the outcome measure the total volume of air displaced per breath?
The outcome measure in my research is TV as the average of 5 relatively even breath with the difference less that 50ml, under natural condition with no intentional control of breath.
Is the measurement of air displacement during inspiration, expiration, or the sum of the two, or something else?
Yes, the amount of air inhaled and exhaled in a single breath without intentional control of breathing.
The authors should state in the manuscript (not just in responses to my comments) why they chose to focus on TV as opposed to the other measurements that the spirometer can make.
→Thank you for your comment.
The promotion of conscious breathing through the cerebral cortex is considered to be at risk for causing hyperpnea and hyperventilation [10]. Therefore, this study uses TV to focus on natural and unintended respiratory changes. I talked about TV and added it to the text.